# A Rho Kinase (ROCK) Inhibitor, Y-27632, Inhibits the Dissociation-Induced Cell Death of Salivary Gland Stem Cells

**DOI:** 10.3390/molecules26092658

**Published:** 2021-05-01

**Authors:** Kichul Kim, Sol Min, Daehwan Kim, Hyewon Kim, Sangho Roh

**Affiliations:** 1Cellular Reprogramming and Embryo Biotechnology Laboratory, Dental Research Institute, Seoul National University School of Dentistry, Seoul 08826, Korea; fpdh0839@naver.com (K.K.); pine93@snu.ac.kr (S.M.); 2Department of Bioengineering and QB3 Institute, University of California, Berkeley, CA 94720, USA; dhpark@berkeley.edu; 3Department of Biomedical Engineering, Hanyang University, Seoul 04763, Korea; khw0063@naver.com

**Keywords:** salivary gland stem cells (SGSCs), ROCK inhibitor, dissociation-induced cell death

## Abstract

Salivary gland stem cells (SGSCs) are potential cell sources for the treatment of salivary gland diseases. The control of cell survival is an essential factor for applying stem cells to regenerative medicine or stem cell-based research. The purpose of this study was to investigate the effects of the ROCK inhibitor Y-27632 on the survival of SGSCs and its underlying mechanisms. SGSCs were isolated from mouse submandibular glands and cultured in suspension. Treatment with Y-27632 restored the viability of SGSCs that was significantly decreased during isolation and the subsequent culture. Y-27632 upregulated the expression of anti-apoptotic protein BCL-2 in SGSCs and, in the apoptosis assay, significantly reduced apoptotic and necrotic cell populations. Matrigel was used to mimic the extracellular environment of an intact salivary gland. The expression of genes regulating apoptosis and the ROCK signaling pathway was significantly reduced when SGSCs were embedded in Matrigel. SGSCs cultured in Matrigel and treated with Y-27632 showed no difference in the total numbers of spheroids and expression levels of apoptosis-regulating genes. Matrigel-embedded SGSCs treated with Y-27632 increased the number of spheroids with budding structures and the expression of acinar cell-specific marker AQP5. We demonstrate the protective effects of Y-27632 against dissociation-induced apoptosis of SGSCs during their culture in vitro.

## 1. Introduction

Saliva plays a crucial role in maintaining oral health, such as oral mucous membrane protection, teeth maintenance, starch digestion, and lubrication. Disruption of the salivary gland by Sjögren’s syndrome or radiation therapy decreases the production and secretion of saliva, which reduces the quality of life [1]. Although saliva substitutes and parasympathetic agonists were developed to lubricate the mouth, benefits are temporary and repeated administration is required [2]. New therapeutic strategies are needed to overcome the limitations of conventional treatment. Experimental approaches using salivary gland stem cells (SGSCs) may provide clues for the development of fundamental treatment methods, such as organ regeneration or replacement.

The identification of cell populations with proliferative and differentiation potential is required for the clinical application of SGSCs. Salivary gland cells expressing c-Kit, CD29, CD24, Ascl3, and EpCAM were previously identified as stem cells [3,4,5,6,7]. In particular, salivary gland cells cultured in vitro as spheroids contained cells expressing the stem cell markers Sca-1, c-Kit, and Musashi-1 and differentiated to ductal and acinar cells [4]. Histological analysis revealed that SOX2-positive cells and terminally differentiated acinar cells could replace damaged secretory cells [8,9]. The transplantation of SGSCs attenuated radiation-induced hyposalivation [4,5,7]. SGSCs with regenerative potential were successfully isolated, yet many in vitro experiments used primary SGSCs without subculture [6,10,11], possibly due to decreased cell survival.

The ROCK signaling pathway, regulated by Rho family GTPases and the downstream effector, ROCK, is an essential process for cellular functions, such as cellular polarity, contractility, motility, proliferation, and apoptosis [12,13]. ROCK signaling is directly related to apoptosis in in vitro cultured cells. Apoptosis in human embryonic stem cells was induced by ROCK-dependent hyperactivation of actin-myosin contraction, and was reduced by knockdown of ROCK1/2 and by treatment with the ROCK inhibitor Y-27632 [14,15]. Y-27632 also reduces apoptosis of human keratinocytes [16], mouse intestinal stem cells [17], and mouse prostate stem cells. Thus, ROCK-mediated dissociation-induced apoptosis is frequently observed in cells with an epithelial phenotype [18]. In SGSCs, however, the cellular effects of the ROCK signaling pathway are rarely reported. Therefore, we present the impact of inhibiting the ROCK signaling pathway on the survival of SGSCs and underlying molecular changes.

## 2. Results

### 2.1. Y-27632 Enhances the Viability of SGSCs

The viability of SGSCs was spontaneously decreased after isolation from the salivary glands and during each subculture. In the passage (P) 2 SGSCs, viability was significantly decreased compared with P0 SGSCs (Figure 1a,b). Five kinase inhibitors (PD184352, PD0325901, SU5402, CHIR99021, and Y-27632) were used for the treatment of SGSCs in vitro. These kinase inhibitors were previously reported to support the maintenance of pluripotent stem cells [19,20,21,22,23,24]. We treated P0 SGSCs with each of the five small molecules for five days to investigate their effects on the formation of spheroids and viability (Figure 2a,b). PD184352 (0.8 μM) and PD0325901 (1 μM), MAPK/Erk kinase 1/2 (MEK1/2) inhibitors, attenuated the formation of spheroids and significantly decreased cell viability. SU5402 (2 μM), an FGF receptor tyrosine kinase inhibitor, also inhibited the formation of spheroids when compared with the control cells; however, it did not affect the viability. Conversely, CHIR99021 (3 μM), a GSK3β inhibitor, enhanced the viability of SGSCs while the morphology of the spheroids was not different from the control cells.

Interestingly, the addition of the ROCK inhibitor Y-27632 (10 μM) increased the spheroid size and viability of SGSCs. Treatment with 10 μM Y-27632 was non-toxic (Figure 2c). The following experiments were conducted using 10 μM Y-27632. SGSCs of each passage were cultured in a medium either with or without Y-27632. In all passages (P0–P3), Y-27632 increased spheroid size (Figure 2d) and improved viability (Figure 2e). These results indicate that the ROCK inhibitor Y-27632 improves the in vitro culture conditions of SGSCs by enhancing viability.

### 2.2. ROCK Inhibitor Inhibits Dissociation-Induced Cell Death in SGSCs

The mRNA expression levels of apoptosis-regulating genes were compared after subculture with or without Y-27632 (Figure 3a). No significant differences were observed in the expression levels of apoptotic genes (*p53*, *Bad*, and *Bax*) or an anti-apoptotic gene (*Bcl-2*) between the control and treatment groups of P0 and P1 SGSCs. The mRNA expression level of *Bcl-2*, however, significantly upregulated in P2 SGSCs. Increased expression of the anti-apoptotic protein, BCL-2, was confirmed in P2 SGSCs after treatment with Y-27632 (Figure 3b). The protein quantification data showed that the expression of BCL-2 was significantly increased by the treatment with Y-27632 (Figure 3c). Apoptosis assay using Annexin V staining also showed that more apoptotic and necrotic cells were present in the control group. Y-27632 significantly reduced early apoptosis (0.32 ± 0.29% versus 1.86 ± 0.97% in the control group), late apoptosis (0.72 ± 0.54% versus 4.43 ± 1.25% in the control group), and necrosis (2.43 ± 1.64% versus 10.43 ± 4.43% in the control group) in SGSCs (Figure 3d,e). These results indicate that Y-27632 enhances the viability of SGSCs by inhibiting apoptosis and necrosis.

### 2.3. Cell–ECM Interactions Suppress Expression of Genes Regulating Apoptosis and the ROCK Signaling Pathway

Apoptosis might be induced by the dissociation of cells from their original environment, such as ECM surrounding salivary gland cells. Thus, SGSCs were embedded into Matrigel to artificially form cell–ECM interactions. The expression levels of apoptosis-regulating genes were measured in SGSCs cultured in suspension and SGSCs embedded in Matrigel. A significant decrease in mRNA expression of *p53*, *Bad*, *Bax*, and *Bcl-2* was observed in Matrigel-embedded SGSCs (Figure 4a). The expression levels of ROCK signaling pathway-related genes (*Rhoa*, *Rhob*, *Rhoc*, *Rock1*, and *Rock2*) were significantly downregulated in Matrigel-embedded SGSCs (Figure 4b). In Matrigel-embedded SGSCs, differences in the size and morphology of the spheroids were observed (Figure 4c), but no significant change was observed in the total number of spheroids between the control and treated cells (Figure 4d). The expression of *Bcl-2*, which was significantly increased by Y-27632 in suspension-cultured SGSCs, was not increased after treatment of Matrigel-embedded SGSCs with Y-27632 (Figure 4e). These results indicate that the disruption of proper cell–ECM interactions might induce cell death by activating the ROCK signaling pathway.

### 2.4. Treatment with Y-27632 Induces Cellular Outgrowth of SGSCs

Because treatment with Y-27632 enhanced the survival of SGSCs by increasing the expression of Bcl-2 and reducing apoptosis and necrosis (Figure 3), we hypothesized that regulation of the ROCK signaling pathway might also be important for the maintenance of the salivary gland. SGSCs were embedded in Matrigel to mimic the in vivo environment of the salivary gland. The morphological and molecular changes associated with the differentiation of embedded SGSCs were assessed. Interestingly, Y-27632 induced cellular outgrowth of SGSCs within six days (Figure 5a), and by the eighth day, Y-27632 significantly increased the number of SGSC-derived spheroids that underwent budding compared with the control group (6.73 ± 3.32% and 25.85 ± 3.74%) (Figure 5b). The expression of *Aqp5* was significantly upregulated in spheroids of the treated cells (Figure 5c). *Mist1* also displayed a significant increase, but only a moderate change was observed. The mRNA expression levels of the functional marker, *Amy1*, and ductal cell marker, *Krt8*, showed no significant difference between the two groups of cells (Figure 5c). Amylase activity was measured in the conditioned medium and the spheroids. Y-27632 did not affect amylase activity as also observed in the qRT-PCR results for *Amy1* (Figure 5d). The protein expression level of AQP5 was significantly upregulated in the spheroids with Y-27632. This was consistent with gene expression data (Figure 5e,f). The protein expression level of KRT8 showed a moderate increase in treated cells, although significance was not observed (Figure 5e,f). These results suggest that Y-27632 induces the differentiation of SGSCs into an acinar cell lineage in a Matrigel-based three-dimensional culture system.

## 3. Discussion

Maintaining adult stem cells in an intact state is essential for proper stem cell preparation and subsequent scientific experiments or therapeutic applications. In this study, five kinase inhibitors were selected and treated for the maintenance of salivary gland stem cells (SGSCs) in vitro. The kinase inhibitors (PD184352, PD0325901, SU5402, CHIR99021, and Y-27632) used in this study were previously reported to support the maintenance of pluripotent stem cells [19,20,21,22,23,24]. Treatment with Y-27632 showed the most significant effects on the morphology and viability of passage 0 (P0) SGSCs. Y-27632 also significantly increased the viability of SGSCs during the entire culture period (P0–P3). These results suggest that inhibition of the Rho kinase (ROCK) signaling pathway prevents the reduction of viability in SGSCs during the isolation and subculture.

Activation of the ROCK signaling pathway is involved in the progression of many diseases such as glaucoma, idiopathic pulmonary fibrosis, chronic kidney diseases, and psoriasis [25,26,27]. In normal conditions, apoptosis works to preserve a biological equilibrium in response to intracellular or extracellular stresses [28]. However, the activation of the ROCK signaling pathway in lung epithelial cells increased the susceptibility to apoptosis after lung injury resulting in abnormal wound healing responses [29,30,31,32]. Therefore, proper inhibition of the ROCK signaling pathway is expected to be important for organ maintenance. In this work, it was hypothesized that the inhibition of the ROCK signaling pathway might promote the survival of SGSCs by inhibiting cell death induced by the dissociation from its original environment. Interestingly, Y-27632 significantly increased the expression of the anti-apoptotic protein BCL-2. In addition, the apoptosis assay revealed that Y-27632 reduced the apoptotic and necrotic cell populations.

The extracellular matrix (ECM) establishes the microenvironment of cells and provides structural support [33]. Previous studies suggested that ECM influences the survival and apoptosis of several cell types [34,35,36,37]. The loss of cell anchorage induces the unbalance between actin–myosin contraction forces and the opposing forces, leading to the disruption of normal cellular morphology and function [14,38,39,40]. If an unbalanced state is maintained, this stress can lead to cell death [41]. In this study, the effect of cell–ECM interactions on the apoptosis of SGSCs was demonstrated. To generate cell–ECM interactions, SGSCs were embedded in Matrigel, and the expression levels of apoptosis-regulating genes were measured. Interestingly, the expression levels of genes regulating apoptosis and the ROCK signaling pathway were significantly reduced in Matrigel-embedded SGSCs compared with suspension-cultured SGSCs, indicating that the dissociation-induced apoptosis was suppressed by the generation of cell–ECM interactions. SGSCs embedded in Matrigel showed downregulated expression of *Rock1* and *Rock2*. The reduction rate of *Rock2*, however, was lower than that of *Rock1*. It was reported that ROCK1 deficiency improves the viability and attachment of mouse embryonic fibroblasts [42]. During apoptosis, the activation of ROCK1 by caspase-mediated cleavage was induced in mouse fibroblasts, while ROCK2 showed no detectable cleavage [43]. According to these previous studies, our results indicate that the rate of downregulation in *Rock1*, which is higher than *Rock2*, may be the result of the attachment of cells on ECM and reduced apoptosis. The effects of Y-27632 on the SGSCs embedded in the Matrigel were also demonstrated. In the previous study, Y-27632 increased the total number of spheroids in suspension-cultured SGSCs [44]. However, in this report, the treatment with Y-27632 of SGSCs that were embedded in Matrigel did not affect the total number of spheroids and the expression of apoptosis-regulating genes. As a regulator of actin–myosin contraction [45], Y-27632 may suppress the dissociation-induced cell death by complementing the cell–ECM interactions.

It has been reported that Y-27632 increased the population of proacinar cells in salivary gland organoids [10]. In SGSCs with high EpCAM expression, the treatment with Wnt3a, Rspo1, and Y-27632 formed organoids with lobular structures and enabled the long-term in vitro culture of SGSCs [7]. In colon cancer cells, the inactivation of RHOA accelerated intestinal tumorigenesis through enhancing the Wnt/β-catenin signaling pathway showing the involvement of Rho GTPase in the Wnt signaling pathway [46]. In this study, expression of the acinar cell marker AQP5 was significantly increased by the treatment with Y-27632, indicating that Y-27632 contributed to the differentiation of SGSCs toward the acinar cell lineage. In addition, treatment with Y-27632 increased the number of SGSC-derived spheroids with budding structures in the absence of Wnt3a and Rspo1, suggesting that Y-27632 may activate the Wnt signaling pathway.

Salivary gland epithelial cell-derived spheroids cultured with Y-27632 increased the expression levels of the salivary gland stem cell marker c-Kit [10], and the transplantation of c-Kit-positive cells restored the tissue homeostasis in an irradiated salivary gland [11]. In addition, the survival of induced pluripotent cell-derived retinal cells transplanted into subretinal space was promoted by Y-27632 [47]. In this study, Y-27632 promoted not only the survival but also the differentiation of SGSCs. Therefore, these results indicate that SGSCs cultured in the presence of Y-27632 may show a more proliferative and adoptive property after transplantation than untreated SGSCs.

In conclusion, we demonstrated the role of the ROCK signaling pathway in the survival of mouse SGSCs. By inhibiting the ROCK signaling pathway through the chemical inhibitor Y-27632, dissociation-induced cell death of SGSCs was significantly reduced. In addition, ECM-based experiments revealed that the disruption of the cell–ECM interactions is associated with increased cell death of SGSCs.

## 4. Materials and Methods

### 4.1. Animals

Five-week-old female mice (C57BL6×DBA2 F1-hybrid) were used. All animal experiments were carried out following the guidelines of the Institutional Animal Care and Use Committee of Seoul National University (Approval number: SNU-180403-4).

### 4.2. Primary Isolation and Culture of SGSCs

Isolated submandibular glands were dissected with scissors in Hanks’ Balanced Salt Solution with CaCl_2_, MgSO_4_ (Welgene, Gyeongsan, Korea) with 1 mg/mL hyaluronidase (Sigma, Louis, MO, USA) and 0.2% collagenase II (Thermo Fisher, Waltham, MA, USA). Mechanically and enzymatically dissociated cells were incubated in 60 mm dishes at 37 °C for 1 h. Cells neutralized with Dulbecco’s modified Eagle’s medium (DMEM; Thermo Fisher) supplemented with 10% fetal bovine serum (FBS; Hyclone Laboratories, Logan, UT, USA), and centrifuged at 500× *g* for 3 min. Centrifuged cells were then suspended in a culture medium composed of DMEM/F12 1:1 (*v/v*; Thermo Fisher), 20 ng/mL fibroblast growth factor-2 (Peprotech, Rocky Hill, NJ, USA), 20 ng/mL epithelial growth factor (Peprotech), 1% N-2 supplement (Thermo Fisher), 1% insulin (Cell application, San Diego, CA, USA), 1 μM dexamethasone (Sigma), and 1% penicillin-streptomycin (Sigma) and filtered with 100 and 40 μM strainers for single-cell isolation. Cells were seeded at 2.0 × 10^5^ cells/mL in poly (2-hydroxyethyl methacrylate) (poly-HEMA; Sigma)-coated 100 mm culture dishes. The spheroids for the subculture were centrifuged at 500× *g* for 3 min and dissociated with 1 mg/mL hyaluronidase and 0.2% collagenase II. After incubation at 37 °C for 30 min, cells were treated with 0.05% trypsin (Thermo Fisher) and isolated into single cells using a 26G syringe needle. Cells were then seeded onto poly-HEMA-coated 6-well plates or 100 mm culture dishes as passage (P) 0 SGSCs and cultured in a CO_2_ incubator (5% CO_2_ in humidified air) for five days, then subcultured with 0.05% trypsin (Thermo Fisher) and labeled as P1 SGSCs.

### 4.3. Viability Assay

A water-soluble tetrazolium salt 1 (WST-1)-based EZ-Cytox kit (Daeil Lab Service, Seoul, Korea) was used for determining cell viability. SGSCs were cultured in 10 μM Y-27632, 0.8 μM PD184352, 1 μM PD0325901, 2 μM of SU5402, and 3 μM CHIR99021 in a 100 mm culture dish for three days, and they were then harvested and transferred into a 96-well plate in a fresh medium. The WST-1 reagent was added to each well and then, the 96-well plates were incubated at 37 °C for 30 min. The absorbance of each well was measured with an Emax Plus Microplate reader at 450 nm. (Molecular Devices, CA, USA). All small molecules were purchased from Selleckchem (Houston, TX, USA).

### 4.4. Cytotoxicity Assay

CytoTox-Glo^TM^ Cytotoxicity Assay (Promega, Fitchburg, WI, USA) was used to assess the cytotoxicity of Y-27632. Y-27632-treated SGSCs were cultured in 96-well plates for three days. Cytotoxicity assay solution (50 μL) was added to each well. The solution was mixed by orbital shaking, and plates were incubated at room temperature for 15 min. Luminescence was measured with a TECAN Spark 10M microplate reader (Tecan, Mannedork, Switzerland).

### 4.5. RNA Extraction, cDNA Synthesis, and qRT-PCR

Total RNA was extracted using a PureLink^TM^ RNA Mini Kit (Invitrogen, Camarillo, CA, USA). cDNA synthesis used M-MLV reverse transcriptase following the manufacturer’s protocol (M.biotech, Hanam, Korea). Real-time PCR was performed using SYBR Pre-mix Ex Taq™ II (Takara, Tokyo, Japan) and a StepOnePlus Real-Time PCR System (Applied Biosystems, Carlsbad, CA, USA). Cycling was performed for 30 s at 95 °C, followed by 40 amplification cycles of 5 s at 95 °C and 30 s at 60 °C. *Gapdh* was used for normalization. Primers are listed in Appendix A.

### 4.6. Western Blot Analysis

Total protein was extracted using Cell Culture Lysis 5× Reagent (Promega) on ice. The protein concentration was measured using a BCA protein assay kit (Thermo Fisher). In total, 20μg of proteins from each group were separated by SDS-PAGE and transferred to polyvinylidene fluoride membranes (Millipore, Bedford, MA, USA). The membranes were blocked with 10% skim milk or 5% BSA for 1 h at room temperature and incubated with primary antibodies overnight at 4 °C. The primary and secondary antibodies were purchased from Bioss (Woburn, MA, USA), Novus biologicals (Littleton, CO, USA), and GeneTex (Irvine, CA, USA). A detailed description of antibodies is listed in the Appendix A. The membranes were washed three times with TBS-T (Tris-buffered saline, 0.1% Tween 20) buffer and incubated with secondary antibodies. The proteins were visualized with an EZ-Western Lumi La (Daeil Lab Service).

### 4.7. Apoptosis Assay by Flow Cytometry

To analyze apoptosis, an Annexin V apoptosis detection kit (Thermo Scientific) was used according to the manufacturer’s instructions. In brief, the cells were harvested and washed once in PBS, then once in 1× Annexin V binding buffer. The cells were resuspended in 100 μL of 1× Annexin V binding buffer and 5 μL of Annexin V was added. The cells were incubated for 15 min at room temperature in the dark. The cells stained with Annexin V were washed once in 1× Annexin V binding buffer and resuspended in 200 μL of 1× Annexin V binding buffer. Then, 5 μL of 7-AAD viability staining Solution was added and analyzed by flow cytometry.

### 4.8. In Vitro Culture of SGSCs in Matrigel

SGSCs were harvested and prepared in a single cell suspension of 1.6 × 10^5^ cells/mL. A total of 25 μL of this suspension was mixed with 50 μL of growth factor-reduced Matrigel (Corning, NY, USA) and deposited in the center of 48-well culture plates. After solidifying the gels at 37 °C for 20 min, the culture medium was added, and SGSCs were cultured for eight days in a CO_2_ incubator (5% CO_2_ in humidified air). A total of 1 mg/mL of dispase (Life Technologies, Camarillo, CA, USA) was added and incubated at 37 °C for 30 min to harvest the SGSCs. The percentage of spheroids with budding structures was measured by counting the number of spheroids with budding structures among the total number of spheroids.

### 4.9. Amylase Activity Assay

Amylase activity was measured using an amylase assay kit (Abcam, Cambridge, MA, USA) following the manufacturer’s instructions. In brief, the harvested spheroids were washed once with cold PBS and resuspended in 500 μL assay buffer. The resuspended and homogenized cells were centrifuged at 4 °C for 5 min at top speed to remove insoluble material and collect the supernatant. The amylase activity of the conditioned medium was tested by direct addition to microplate wells.

### 4.10. Statistical Analysis

All data were presented as the mean ± S.D., and Student’s *t*-test, as well as one-way ANOVA followed by Tukey’s post-hoc test, was performed to determine the significance of experimental differences. Statistical analyses were performed using GraphPad Prism 5.0. All experiments were independently repeated at least three times. *p* < 0.05 was considered statistically significant.

## Figures and Tables

**Figure 1 molecules-26-02658-f001:**
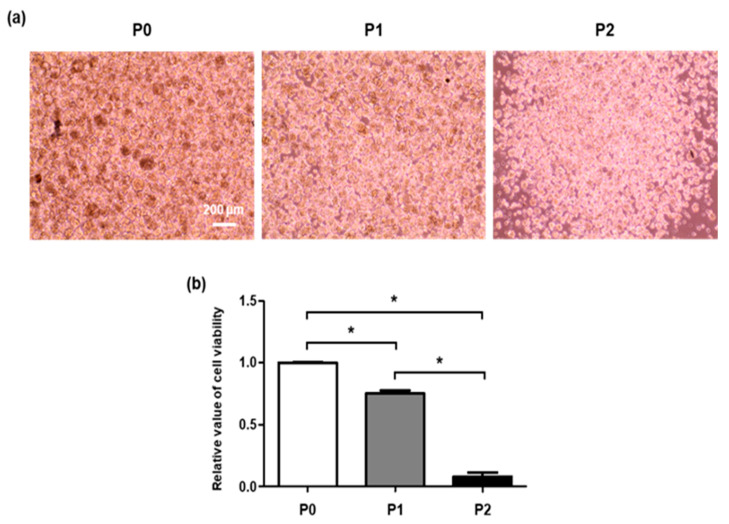
Decreased viability of SGSCs after dissociation. (**a**) Morphology of SGSC-derived spheroids in passage 0–2 (P0, P1, and P2). Scale bar = 200 μM. (**b**) Relative value of viability was measured in P0, P1, and P2 SGSCs. * *p* < 0.05.

**Figure 2 molecules-26-02658-f002:**
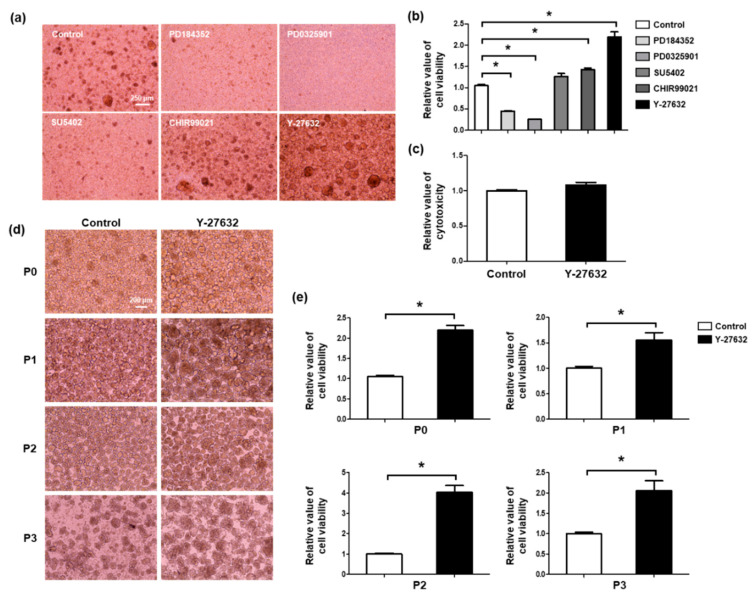
Search for a kinase inhibitor(s) that enhances the viability of SGSCs. (**a**) Morphology of spheroids after the treatment with kinase inhibitors (PD184352, PD0325901, SU5402, CHIR99021, and Y-27632) for five days. Scale bar = 250 μM. (**b**) Relative value of viability after the treatment with kinase inhibitors for five days. (**c**) Relative value of cytotoxicity measured after treatment with Y-27632 for five days. (**d**) Morphology of SGSCs in each passage (P0–P3) treated with Y-27632. Scale bar = 200 μM. (**e**) Relative value of viability was measured in P0, P1, P2, and P3 SGSCs after treatment with Y-27632. * *p* < 0.05.

**Figure 3 molecules-26-02658-f003:**
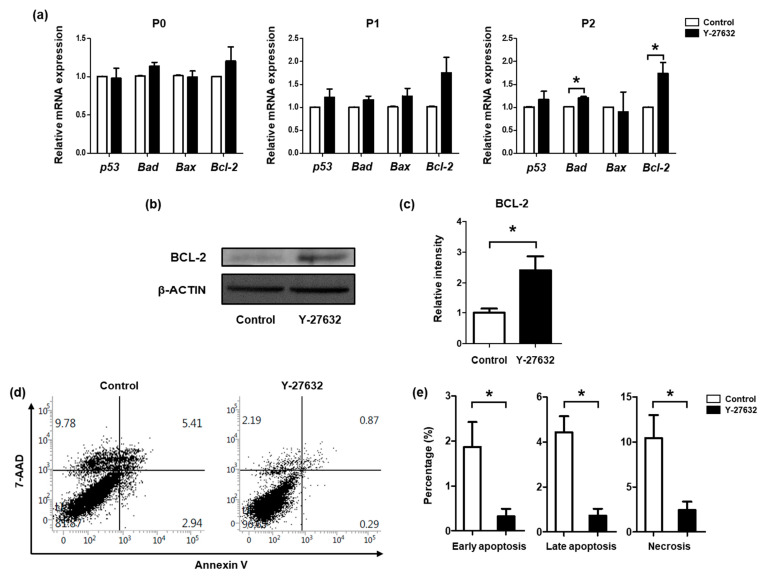
Anti-apoptotic potential of Y-27632. (**a**) Expression levels of apoptotic and anti-apoptotic genes were analyzed in P0, P1, and P2 SGSCs using qRT-PCR. (**b**) Western blotting analysis was used to confirm the protein expression level of BCL-2 in P2 SGSCs. (**c**) Quantification data of western blotting results shown in panel b. (**d**) Apoptotic cell populations in P2 SGSCs were analyzed using Annexin V and 7-AAD staining. (**e**) Percentages of early apoptotic, late apoptotic, and necrotic cell populations in P2 SGSCs are represented as bar graphs. * *p* < 0.05.

**Figure 4 molecules-26-02658-f004:**
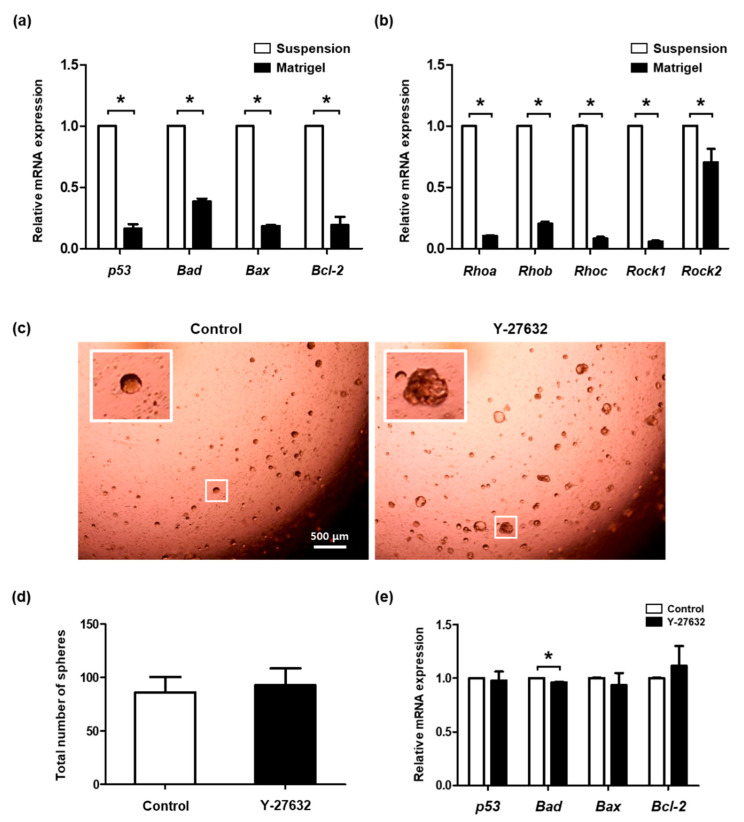
Effects of Y-27632 on the Matrigel-embedded SGSCs. (**a**,**b**) Expression levels of genes regulating apoptosis and the ROCK signaling pathway were compared between suspension-cultured and Matrigel-embedded SGSCs. (**c**) Morphology of SGSCs embedded in Matrigel and cultured for five days in the absence and presence of Y-27632. Scale bar = 500 μM. (**d**) The total number of spheroids was measured and compared between the control and treated cells. (**e**) Expression levels of apoptosis-regulating genes were assessed. * *p* < 0.05.

**Figure 5 molecules-26-02658-f005:**
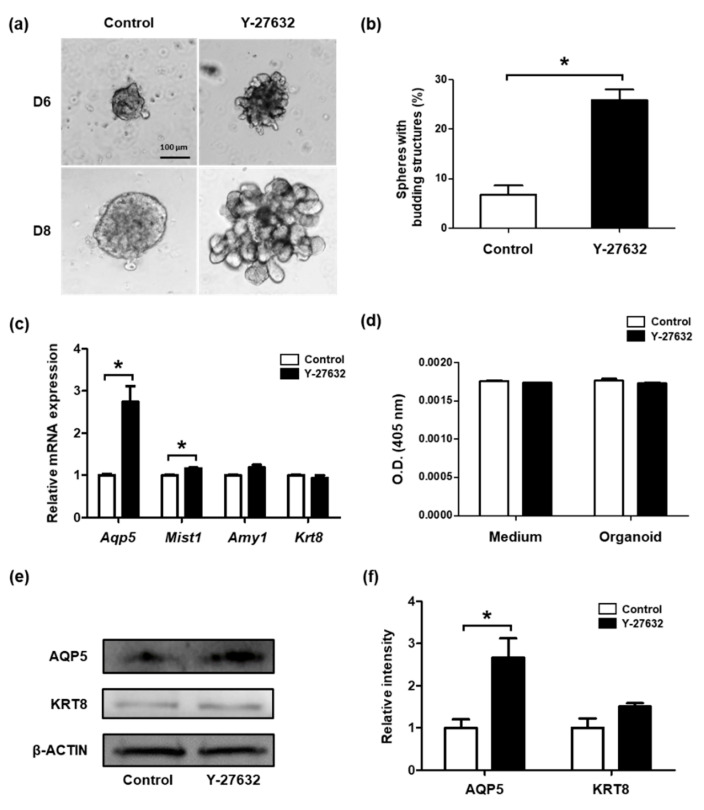
Effects of Y-27632 on the generation of salivary gland-like structures. (**a**) Morphology of SGSC-derived spheroids generated after embedding in Matrigel in the presence and absence of Y-27632. Images were obtained on days six and eight after embedding. Scale bar = 100 μM. (**b**) Proportion of spheroids with budding structures. (**c**) Expression of acinar- and duct-specific markers. Their mRNA levels were measured using qRT-PCR. (**d**) Amylase activity in the conditioned medium and SGSCs. (**e**) Western blotting analysis confirming protein expression levels of the acinar and ductal cell markers. (**f**) Quantification of western blotting results shown in panel e. * *p* < 0.05.

## Data Availability

All the data in this report are presented in manuscript.

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
