# Peer review of "A Rho Kinase (ROCK) Inhibitor, Y-27632, Inhibits the Dissociation-Induced Cell Death of Salivary Gland Stem Cells"

_molecules, 2021, doi:10.3390/molecules26092658_

Round 1

Reviewer 1 Report

In this work, the authors obtained a preparation containing salivary gland stem cells (SGSCs), which could be cultured in suspension and in Matrigel. Because the Rho kinase (ROCK) inhibitor Y-27632 has been reported to reduce apoptosis of several cells, they here studied the effects of Y-27632 on SGSCs under various experimental conditions. In this manuscript, the authors report that Y-27632 inhibits dissociation-induced cell death of SGSCs, and that the disruption of the cell-extracellular matrix (ECM) interactions is associated with increased death of SGSCs. This manuscript thus contains useful information for future work on SGSCs.  

Major comment:

              The authors describe the method for obtaining the preparation containing SGSCs in the subsection "4.2. Primary isolation and culture of SGSCs"; however, information on the characteristics of the SGSC preparation, such as cellular composition, is not provided. Such information might be required by readers for correct understanding of the reported results. It would be preferable to provide such information in an appropriate part of the text or as supplementary information. In addition, authors use the expression "P0 SGSCs" (e.g., p. 2, line 62). It would also be helpful to clearly define the preparation corresponding to "P0 SGSCs", probably in the subsection 4.2 on p. 8.

Minor comments:

During the review, I noticed several points to be considered by the authors. These are listed below (comments/suggestions are indicated by "-->"):

p. 1, line 2:

              Rho kinase (ROCK) inhibitor, Y-27632,

-->        A Rho kinase (ROCK) inhibitor, Y-27632,

p. 1, line 15:

              ROCK inhibitor Y-27632

-->        the ROCK inhibitor Y-27632

p. 1, line 17:

              after isolation and subsequent culture.

-->        during isolation and subsequent culture.

p. 1, lines 18-19:

              Y-27632 upregulated the expression of anti-apoptotic protein Bcl-2 in SGSCs and in apoptosis assay, Y-27632 significantly reduced apoptotic and necrotic cell populations.

-->        Y-27632 upregulated the expression of anti-apoptotic protein Bcl-2 in SGSCs andin apoptosis assay, Y-27632 significantly reduced apoptotic and necrotic cell populations.

p. 1, lines 19-20:

              Matrigel was used to mimic intact salivary gland and investigate the effects on the survival of SGSCs.

-->         Matrigel was used to mimic the extracellular environment of intact salivary gland and investigate the effects on the survival of to keep viability of SGSCs.

p. 1, lines 20-21:

              Expression of apoptosis and ROCK signaling pathway regulating genes significantly reduced

-->?      Expression of genes regulating apoptosis and ROCK signaling pathway regulating genes was significantly reduced

p. 1, line 22:

              SGSCs in Matrigel and treated with Y-27632

-->        SGSCs cultured in Matrigel and treated with Y-27632

p. 1, line 23:

              apoptosis regulating genes.

-->        apoptosis-regulating genes.

p. 1, line 26:

              SGSCs cultured in vitro

-->        SGSCs during their culture in vitro

p. 1, lines 32-33:

              hypo salivation and decreased production and secretion of saliva reduces

-->        hypo salivation, and decreased production and secretion of saliva reduces

p. 1, line 35:

              but benefits are temporary

-->        but benefits are temporary

p. 1, line37:

              SGSCs

-->        salivary gland stem cells (SGSCs)

p. 2, lines 47-48:

              pathway, regulated by Rho family GTPases and downstream effector, ROCK, is an essential mediator of cellular functions,

-->        pathway, regulated by Rho family GTPases and the downstream effector ROCK, is an essential mediator of process for cellular functions,

p. 2, lines 51-52:

              reduced by knockdown of ROCK1/2 and Y-27632 treatment [14,15].

-->        reduced by knockdown of ROCK1/2 and by treatment with the ROCK inhibitor Y-27632 treatment [14,15].

p. 2, line 55:

              the effects of ROCK signaling pathway

-->        the cellular effects of ROCK signaling pathway

p. 2, lines 63-64:

              were selected and treated for maintenance of SGSCs in vitro.

-->        were selected and treated used for maintenance treatment of SGSCs in vitro.

p. 2, lines 65-67:

              One of five small molecules was treated to P0 SGSCs cells for five days to investigate the effects of small molecule on the formation of ...

-->        We treated P0 SGSCs with each of the One of five small molecules was treated to P0 SGSCs cells for five days to investigate the their effects of small molecule on the formation of ...

p. 2, lines 67-68:

              Extracellular-signal regulated kinase (ERK) 1/2 inhibitors (0.8 μM PD184352 and 1 μM PD0325901) attenuated ...

-->        PD184352 (0.8 μM) and PD0325901 (1 μM), MAPK/Erk kinase 1/2 (MEK1/2) inhibitors, attenuated

p. 2, lines 69-70:

              FGF receptor tyrosine kinase inhibitor (2 μM SU5402) also inhibited ...

-->        SU5402 (2 μM), an FGF receptor tyrosine kinase inhibitor, (2 μM SU5402) also inhibited ...

p. 2, lines 71-72:

              Conversely, GSK3β inhibitor (3 μM CHIR99021) enhanced viability of SGSCs and morphology of spheres was not different ...

-->        Conversely, CHIR99021 (3 μM), a GSK3β inhibitor(3 μM CHIR99021) enhanced viability of SGSCsand while morphology of spheres was not different ...

p. 2, line 73:

              Interestingly, addition of ROCK inhibitor (10 μM Y-27632) increased ...

-->        Interestingly, addition of the ROCK inhibitor Y-27632 (10 μM) (10 μM Y-27632) increased ...

p. 2, line 78:

              that ROCK inhibitor Y-27632

-->        that the ROCK inhibitor Y-27632

p. 2, line 81:

              Expression levels of apoptosis regulating genes

-->        The mRNA expression levels of apoptosis-regulating genes

p. 2, lines 81-82:

              after treatment with Y-27632

-->        after subculture with and without Y-27632

p. 2, line 83:

              and anti-apoptotic gene (Bcl-2)

-->        and an anti-apoptotic gene (Bcl-2)

p. 2, line 84:

              mRNA expression level of Bcl-2

-->        The mRNA expression level of Bcl-2

p. 2, lines 84-85:

              mRNA expression level of Bcl-2 increased in P0, P1, and P2 SGSCs, significance upregulation was observed ...

-->        The mRNA expression level of Bcl-2 was increased by Y-27632 in P0, P1, and P2 SGSCs; however, significant upregulation was observed ...

p. 2, line 86:

              Increased expression of anti-apoptotic protein, Bcl-2, was

-->        Increased expression of the anti-apoptotic protein Bcl-2 was

p. 2, lines 87-88:

              expression of Bcl-2 significantly increased after treatment with Y-27632

              expression of Bcl-2 was significantly increased by the treatment with Y-27632

p. 2, line 89:

              showed more apoptotic and necrotic cells are present

-->        showed that more apoptotic and necrotic cells were present

p. 2, lines 90-91:

              (1.86 ± 0.97% and 0.32 ± 0.29%)

-->        (1.86 ± 0.97% and 0.32 ± 0.29% versus 1.86 ± 0.97% in control)

p. 2, line 91:

              (4.43 ± 1.25% and 0.72 ± 0.54%)

-->        (4.43 ± 1.25% and 0.72 ± 0.54% versus 4.43 ± 1.25% in control)

p. 2, lines 91-92:

              (10.43 ± 4.43% and 2.43 ± 1.64%)

-->        (10.43 ± 4.43% and 2.43 ± 1.64% versus 10.43 ± 4.43% in control)

p. 3, lines 96-97:

              Morphology of SGSC-derived 96 spheres (P0, P1, and P2).

-->        Morphology of SGSC-derived 96 spheres in passages 0-2 (P0, P1, and P2).

p. 3, line 98, Figure 2e:

-->        Comment: 'P0', 'P1', 'P2', and 'P3' should not be included in the titles of the vertical axes; they should be placed in other appropriate places.

p. 3, line 100:

              Search for kinase inhibitor that enhances

-->        Search for a kinase inhibitor(s) that enhances

p. 3, line 101:

              treatment of kinase inhibitors (PD184352, ...

-->        treatment of with kinase inhibitors (PD184352, ...

p. 3, line 102:

              after the treatment of small molecules.

-->?      after the treatment of small molecules with kinase inhibitors for five days.

p. 3, line 103:

              cytotoxicity measured after treatment with Y-27632.

-->?      cytotoxicity measured after treatment with Y-27632 for five days.

p. 3, line 103:

              (d) Morphology of SGSCs in each passage after treatment with

-->        (d) Morphology of SGSCs in each passage (P0-P3) treated with

p. 3, line 105:

              after treatment with

-->        treated with

p. 4, line 107:

              apoptotic and anti-apoptotic gene

-->        apoptotic and anti-apoptotic genes

p. 4, lines 108-109:

              to confirm protein expression level of Bcl-2 in P2 SGSCs.

-->        to confirm the protein expression level of Bcl-2 in P2 SGSCs.

p. 4, line 109:

              Quantification data of western blotting results.

-->        Quantification data of western blotting results shown in panel b.

p. 4, lines 109-110:

              Apoptotic cell populations were analyzed

-->?      Apoptotic cell populations in P2 SGSCs were analyzed

p. 4, line 111:

              apoptotic, and necrotic cell populations

-->?      apoptotic, and necrotic cell populations in P2 SGSCs

p. 4, lines 112-113:

              suppress expression of apoptosis and ROCK signaling pathway regulating genes

-->?      suppress expression of genes regulating apoptosis and ROCK signaling pathway regulating genes

p. 4, line 114:

              induced because of dissociation of cells

-->        induced because of by dissociation of cells

p. 4, line 117:

              apoptosis regulating genes

-->        apoptosis-regulating genes

p. 4, lines 117-118:

              measured between SGSCs cultured in suspension and SGSCs embedded in Matrigel.

-->        measured between in SGSCs cultured in suspension and in SGSCs embedded in Matrigel.

p. 4, lines 121-122:

              Matrigel-embedded SGSCs treated with Y-27632 showed different results from SGSCs cultured in suspension.

-->        Comment/Question: Were the control SGSCs cultured in suspension? It seems likely that they were embedded in Matrigel and cultured in the absence of Y-27632.

p. 4, line 123:

              (Figure 4b)

-->        (Figure 4c)

p. 4, line 125:

              (Figure 4c)

-->        (Figure 4d)

p. 4, lines 126-127:

              did not increase after treatment with Y-27632 (Figure 4d).

-->?      was not increased after treatment of Matrigel-embedded SGSCs with Y-27632 (Figure 4e)

p. 5, lines 130-131:

              Expression levels of apoptosis and ROCK signaling pathway regulating genes were ...

-->        Expression levels of genes regulating apoptosis and ROCK signaling pathway regulating genes were ...

p. 5, lines 132-133:

              Morphology of SGSCs embedded in Matrigel and cultured for five days.

-->?      Morphology of SGSCs embedded in Matrigel and cultured for five days in the absence and presence of Y-27632.

p. 5, line 135:

              apoptosis regulating genes

-->        apoptosis-regulating genes

p. 5, lines 137-138:

              Based on the results that treatment with Y-27632 enhanced the survival of SGSCs by increasing the expression of Bcl-2 and reducing apoptosis and necrosis (Figure 4),

-->?      Based on the results that Because treatment with Y-27632 enhanced the survival of SGSCs by increasing the expression of Bcl-2 and reducing apoptosis and necrosis (Figure 3),

p. 5, line 139:

              regulation of ROCK signaling pathway might also be important factor

-->        regulation of ROCK signaling pathway might also be important factor

p. 5, line 142:

              with the differentiation of SGSCs were assessed.

-->        with the differentiation of embedded SGSCs were assessed.

p. 5, line 143:

              on the 8th day,

-->        by the 8th day,

p. 6, lines 159-160:

              Morphology of SGSC-derived spheres generated after embedding in Matrigel on day six and eight in the presence and absence of Y-27632.

-->?      Morphology of SGSC-derived spheres generated after embedding in Matrigel on day six and eight in the presence and absence of Y-27632. Images were obtained on days six and eight after embeding.

p. 6, line 161:

              spheres with a budding structures.

-->        spheres with a budding structures.

p. 6, lines 161-162:

              Expression of acinar- and duct-specific markers measured using qRT-PCR.

-->        Expression of acinar- and duct-specific markers. Their mRNA levels were measured using qRT-PCR.

p. 6, lines 162-163:

              Amylase activity measured in conditioned medium and SGSCs.

-->        Amylase activity measured in conditioned medium and SGSCs.

p. 6, line 163:

              Western blotting analysis used to confirm protein expression levels of

-->        Western blotting analysis used to confirming protein expression levels of

p. 6, line 164:

              Quantification of western blotting results.

-->        Quantification of western blotting results shown in panel e.

p. 7, line 171:

              Treatment of Y-27632

-->        Treatment with Y-27632

p. 7, line 175:

              viability in SGSCs after the isolation and subculture.

-->        viability in SGSCs during the isolation and subculture.

p. 7, lines 183-187:

-->        Comment: The description "treatment of Y-27632 showed ...  ... after the isolation and subculture." is essentially the same as the description "Treatment of Y-27632 showed ...  ... after the isolation and subculture (p. 7, lines 171-175)."   

-->?      In this report work, treatment of Y-27632 showed the most significant effects on the viability of SGSCs after primary culture. Y-27632 also significantly increased the viability of SGSCs during the entire culture period (P0-P3). These results suggest that inhibition of the Rho kinase (ROCK) signaling pathway prevents the reduction of viability in SGSCs after the isolation and subculture. Based on the results of in vitro viability assay data, it was hypothesized that the inhibition of the ...

p. 7, line 196:

              opposing forces leading

-->        opposing forces, leading

p. 7, lines 200-201:

              apoptosis regulating genes

-->        apoptosis-regulating genes

p. 7, lines 201-202:

              the expression levels of apoptosis and ROCK signaling pathway regulating genes

-->        the expression levels of genes regulating apoptosis and ROCK signaling pathway regulating genes

p. 7. lines 207-208:

              the treatment of Y-27632 to SGSCs that are embedded in Matrigel

-->        the treatment with Y-27632 of SGSCs that were embedded in Matrigel

p. 7, line 209:

              apoptosis regulating genes.

-->        apoptosis-regulating genes.

p. 7. lines 213-214:

              the treatment of Wnt3a, Rspo1, and Y-27632

-->        the treatment with Wnt3a, Rspo1, and Y-27632

p. 7, line 218:

              expression of acinar cell marker Aqp5

-->        expression of the acinar cell marker Aqp5

p. 7, lines 218-219:

              increased with the treatment of Y-27632,

-->        increased by the treatment with Y-27632,

p. 8, line 223:

              In conclusion, we demonstrate the role of

-->        In conclusion, we demonstrated the role of

p. 8, lines 226- 227:

              the disruption of the cell-ECM interactions are associated with

-->        the disruption of the cell-ECM interactions is associated with

p. 8, line 237:

              in 60 mm dishes

-->        in 60-mm dishes

p. 8, line 239:

              centrifuged at 500 g for 3 min and filtered with

-->?      centrifuged at 500 g for 3 min, suspended in ... , and filtered with

p. 8, line 239:

              100 and 40 mm strainers

-->        100- and 40-mm strainers

p. 8, line 241:

              100 mm culture dishes.

-->        100-mm culture dishes.

p. 8, lines 242-243:

              Dulbecco’s Modified Eagle’s medium/F12 1:1 (v/v; Thermo Fisher), 20 ng/mL fibroblast growth factor-2

-->        Dulbecco’s Modified Eagle’s medium/F12 1:1 (v/v; Thermo Fisher) containing 20 ng/mL fibroblast growth factor-2

p. 8, lines 240-245:

-->        Comment: Culture conditions for isolated cells, such as the culture time period, are not described. It seems likely that cells were cultured in a CO2incubator.

p. 8, line 249:

              Cells were then seeded into poly-HEMA-coated 6-well plates

-->?      Cells were then seeded as SGSCs into poly-HEMA-coated 6-well plates

p. 8, lines 249-250:

              100 mm culture dishes.

-->        100-mm culture dishes.

p. 8, lines 254-255:

              100 mm culture dishes.

-->        100-mm culture dishes.

p. 8, line 257:

              by an Emax Plus Microplate reader

-->        with an Emax Plus Microplate reader

p. 9, lines 281-282:

-->        Comment: The primary and secondary antibodies are not specified; information on them, such as antibody sources, should be provided.

p. 9, line 291:

              5mL of 7-AAD Viability Staining Solution

-->        Five mL of 7-AAD Viability Staining Solution

p. 9, line 298:

              cultured for eight days.

-->?      cultured for eight days in a COincubator (5% COin humidified air).

p. 9, line 303:

              Amylase activity assay was measured

-->        Amylase activity assay was measured

p. 9, line 305:

              Resuspended cells

-->?      Resuspended and homogenized cells

p. 9, line 313:

              *p< 0.05

-->        p< 0.05

Reviewer 2 Report

In this manuscript, Kim et al demonstrated the importance of the ROCK signaling pathway for the survival of mouse salivary gland stem cells (SGSCs), which are a potential cell source for salivary gland diseases, by using the ROCK chemical inhibitor Y-27632. Also, they revealed in the manuscript that the cell death of SGSCs increase by disrupting the cell-extracellular matrix interactions. In this manuscript, appropriate methods have been used and the data are convincing. Although the manuscript is well written, it can be improved by minor corrections suggested below.

  1. Please change ‘spheres’, which mentioned throughout the text, to ‘spheroids’.
  2. In section 4.6 Western blot analysis, please include the details of primary antibodies such as company name, catalog number, dilution. Alternatively, list those details in a supplementary table (Table S2).
  3. Figure 4b and lines 119-120, the downregulation of Rock2 expression levels in matrigel-embedded SGSCs is comparatively low. This variation needs a brief explanation.

Reviewer 3 Report

Rho kinase inhibitor inhibits dissociation induced cell death of salivary gland stem cells

Mouse submandibular gland cultured as suspension cell alone are not viable. Treatment with the inhibitor Y27632 increased viability of the cells.  Bcl-2 was up-regulated by Y27632.  Cell death was inhibited by the Y27632.  Cell death was inhibited by the Y27632.

Cells can also be cultured in matrigel to enhance their survival in vitro.  When the inhibitor was included cell budding was increased by the 8th day and Aqp5 was significantly upregulated, a marker for differentiation.  This implies a more differentiated cell phenotype.

The stated ultimate goal of these studies at the outset was to be able to adoptively transfer stem cells to alleviate disease.  What is lacking here is a demonstration that these cells treated with the inhibitor are more readily accepted and proliferating in an adoptive cell transfer.

Round 2

Reviewer 3 Report

adoptive transfers should be stated as a goal.